# Serum Iron Level and 10-Year Survival after Melanoma

**DOI:** 10.3390/biomedicines10123018

**Published:** 2022-11-23

**Authors:** Karolina Rowińska, Piotr Baszuk, Emilia Rogoża-Janiszewska, Jakub Deptuła, Wojciech Marciniak, Róża Derkacz, Marcin Lener, Cezary Cybulski, Magdalena Kiedrowicz, Magdalena Boer, Mariola Marchlewicz, Tadeusz Dębniak, Jan Lubiński

**Affiliations:** 1Department of Genetics and Pathology, International Hereditary Cancer Center, Pomeranian Medical University, 71-252 Szczecin, Poland; 2Department of Skin Diseases and Venerology PUM, Pomeranian Medical University, 72-010 Police, Poland

**Keywords:** melanoma, iron, survival

## Abstract

The malignant melanoma of the skin is a very aggressive tumor. The determination of prognostic biomarkers is important for the early detection of recurrence, and for the enrollment of the patients into different treatment regimens. An evaluation of a cohort of 375 Polish MM cases revealed that a low serum iron concentration (i.e., below 893.05 µg/L) was associated with increased mortality. The study group was followed up from the date of melanoma diagnosis until death or 2020. Patients were assigned to one of four categories in accordance with increasing iron level (I–IV quarters). Patients with a low iron level of below 893.05 µg/L (I quarter) had a significantly lower survival rate when compared to the subgroup with the highest iron level, above 1348.63 µg/L (IV quarter; HR = 4.12; *p* = 0.028 and HR = 4.66; *p* = 0.019 for uni- and multivariable models, respectively). Multivariable analysis took into account the following factors: iron levels, Clark, sex, and age. Future studies based upon the examination of a larger number of cases should be conducted to confirm our findings.

## 1. Introduction

The malignant melanoma of the skin is a very aggressive tumor. In 2020, around 320,000 cases were globally diagnosed. Its frequency has increased in the last few years by nearly 50%, with deaths increasing by 32%. The WHO predicts that the number of deaths caused by this malignancy will increase by 20% by 2025 and up to 74% by 2040 [1]. 

The 5-year survival rate of melanoma patients is 99% in localized, 66% in regional, and 23% in distant stages. Breslow thickness, Clark, and tumor node and metastasis (TNM) stating are used in clinical prediction for melanoma outcome; however, to more precisely evaluate prognoses of MM cases, new tools have to be developed [2]. 

Iron, the most abundant trace element in the human body, is involved in various biological processes, such as oxygen transport, mitochondrial respiration, and DNA synthesis [3,4]. Iron is redox-active and can cause tissue injury from the excessive generation of reactive oxygen species [5]. The role of oxidative stress in suppressing melanoma metastasis is well-documented [4,6,7,8]. Ferroptosis is a newly identified form of iron-dependent regulated cell death that is morphologically, biochemically, and genetically distinct from apoptosis, autophagy, pyroptosis, and necroptosis [9]. Ineffective ferroptosis is associated with tumorigenesis, and the dysregulation of ferroptosis is linked to the development of breast, cervical, colorectal, gastric, hepatocellular, lung, ovarian, and prostate cancers, renal carcinomas, and melanoma [10]. Melanoma dedifferentiation status is correlated with ferroptosis sensitivity. Talty et al. provided data that supported roles for ferroptosis in melanoma pathogenesis, metastasis, and treatment, such as melanoma dedifferentiation and metastatic potential correlating with ferroptosis sensitivity [11]. The pathways of iron uptake, storage, mobilization, trafficking, and regulation are all perturbed in cancer, suggesting that the reprogramming of iron metabolism is a central aspect of tumor cell survival. Anemia is frequently observed in many patients with cancer, and iron homeostasis dysregulation is implicated in numerous types of cancers [12]. There are conflicting data in terms of serum iron level, its other parameters (e.g., TIBC, UIBC, transferrin saturation, ferritin level), and cancer course (melanoma was not evaluated in these studies). Recently, an association between serum iron concentration and lung cancer survival has been suggested. No studies examining the role of serum iron concentration and melanoma prognosis have been published. Herein, we show the 10-year survival of melanoma patients in relation to their serum iron levels.

## 2. Materials and Methods

### 2.1. Study Participants

A total of 375 melanoma patients were included in the study after providing written informed consent. They were chosen from a record of 1500 MM cases with histopathologically confirmed disease housed at the Hereditary Cancer Center in Szczecin, and diagnosed between 2006 and 2016 in Polish cities (Szczecin, Zielona Gora, Gorzow Wielkopolski, Opole, Bialystok). All newly diagnosed melanoma cases with a secured biobank were enrolled in the study. The research was conducted in accordance with the Declaration of Helsinki, and all participants signed a written informed consent document prior to donating a blood sample for analysis. The research was approved by the Ethics Committee of the Pomeranian Medical University in Szczecin (number KB-0012/73/10, 21 June 2010). All patient blood samples were collected at the time of melanoma diagnosis, but before the commencement of any treatment other than the surgical removal of skin lesions. Consenting patients were asked not to eat at least 4 h before the blood collection. A blood sample (10 cc) was obtained during the diagnostic workup and collected into tubes certified for the quantification of trace metals (Vacutainer® System, royal blue cap). Blood samples were taken between 8 a.m. and 2 p.m., and were centrifuged within 30 to 120 min of collection to separate the serum from the cellular fraction. The serum samples were stored at −80 °C until required for the iron assay.

### 2.2. Measurement of Iron Level

Serum iron levels were quantified with inductively coupled mass spectrometry (ICP-MS, Nex-ION 350D, Perkin Elmer, Shelton, CT, USA) using high-purity ammonia gas for a reduction in polyatomic interferences. Spectrometer was calibrated using a matrix-matched calibration technique. Calibration standards were prepared fresh daily from 10 µg/mL Multi-Element Calibration Standard 3 (PerkinElmer, Shelton, CT, USA) by diluting them with a blank reagent to final concentrations of 10, 20, 50, and 100 µg/l. Correlation coefficients for the calibration curves were always greater than 0.999. Rhodium (105Rh) was set as the internal standard. The analytical protocol assumed a 30-fold dilution of the serum in the blank reagent. The blank reagent consisted of high-purity water (>18 MΩ), TMAH (AlfaAesar), Triton X-100 (PerkinElemer), n-butanol (Merck), and EDTA (Sigma Aldrich, Steinheim, Germany). The accuracy and precision of measurements were tested using certified reference material (CRM), Clincheck Plasmonorm Serum Trace Elements Level 1 (Recipe, Munich, Germany). Technical details, plasma operating settings, and mass-spectrometer acquisition parameters are available on request.

### 2.3. Statistical Analysis

A total of 375 melanoma patients were included in the study after providing their written informed consent. Patients were classified into one of four categories, equal in terms of the number of participants, in accordance with increasing iron level (I–IV quarters). The study groups were followed up from the date of melanoma diagnosis until death or 2020. The range of follow-ups varied between 0 and 10 years (observation time longer than 10 years was considered to be a 10-year observation). Univariable and multivariable Cox proportional hazard models were used in order to calculate the hazard ratios (HRs). Multivariable analysis took into account the following factors: iron levels (divided into quarters), Clark (II, III, IV/V), sex (male/female), and age (continuous variable). Due to the relatively small number of cases with Clark V, the authors decided to include them into the Clark IV category for calculation purposes. Subjects with Clark I were excluded from the database because this type of melanoma does not influence mortality. The survival rates in the univariable approach are graphically represented with Kaplan–Meier curves. Patients with the highest iron levels (IV quarter) were chosen as the reference due to the highest alive/deceased ratio. In order to calculate differences between iron levels between the sexes and Clark factors, the t- or Wilcoxon rank-sum test was applied depending on the normality of the given data distributions. In order to determine the correlation between Breslow thickness and iron level, the linear regression model was used. All calculations were performed using R, a language and environment for statistical computing (R Foundation for Statistical Computing, Vienna, Austria, R version 4.0.4, 10 October 2020).

## 3. Results

The mean age of diagnosis was 54.63 years (range 21–90 years); 62% of the patients were females, and 39% were diagnosed with melanoma Clark Stage IV/V. The characteristics of the subjects, confidence intervals, hazard ratios, and p-values for univariable and multivariable Cox proportional hazard regression models for all analyzed factors are presented in Table 1.

Median Fe level among all melanoma patients was 1105.49 μg/L, mean Fe level was 1133.57 μg/L (range from 162.15 μg/L to 2815.78 μg/L) and the interquartile range (IQR) was 453.32 μg/L. A significant difference in iron concentration was observed in relation to sex (*p* < 0.001). There were no significant differences in iron levels in relation to the Clark categories (*p* > 0.3). There was also no significant correlation between Breslow thickness and iron level (*p* = 0.4). The mean, standard deviation, median, range, and IQR are shown in Table 2.

Patients with a low iron level, below 893.05 μg/L (I quarter), had higher HR compared to the subgroup with the highest iron level, above 1348.63 μg/L (IV quarter; HR = 4.12; *p* = 0.028 and HR = 4.66; *p* = 0.019 for both uni- and multivariable models, respectively). 

Men had higher HR than that of women; however, in the univariable model, the result was on the border of significance (HR 2.09; *p* = 0.042). In the multivariable models, differences between HRs depending on sex were not significant (HR = 1.88; *p* = 0.092).

Hazard ratios were significantly higher with increasing age (HR = 1.06; *p* < 0.001 and HR = 1.05; <0.001 for univariable and multivariable models, respectively). In univariable analysis, we also confirmed the association between Breslow thickness and melanoma outcome (Table 1). 

Melanoma patients with Clark IV/V had higher HR compared to the patients with Clark II (HR = 8.76; *p* = 0.035 and HR = 6.84; *p* = 0.062—in univariable and multivariable models).

The Kaplan–Meier survival curves in relation to the quarters of serum iron levels are presented in Figure 1.

## 4. Discussion

The carcinogenic potential of iron was proven, but evidence from observational studies that have linked serum iron variables and cancer outcomes is inconsistent.

According to Quintana et al., there were no significant overall associations between serum iron, transferrin or TSAT and breast, prostate, lung, and colorectal cancer mortality. Inverse associations between ferritin levels, and breast cancer risk and cancer mortality were found [13].

Several studies suggest an association between higher Fe levels and the better prognosis of cancer patients. Iron deficiency (measured by transferrin saturation—TSAT) was associated with poor outcome in patients with pancreatic, colorectal, and lung cancers [14].

Patients with low preoperative serum iron level had worse postoperative survival and higher recurrence rate in hepatocellular cancer [15].

A higher Fe body content was consistently associated with the better survival of lung-cancer patients [16].

Serum iron was also suggested to be closely related to the overall survival of oral cancer, and a lower risk of death was reported among patients with high serum iron [17].

Both low and high preoperative iron in Stage II–III colorectal patients were associated with unfavorable overall survival in a study conducted by Sawayama et al. [18].

In a Taiwanese study, Wen et al. reported that elevated serum iron concentrations were associated with higher risks of cancer (especially liver and breast) and cancer mortality [19]. In another study of a national cohort of United States adults, Wu et al. suggested an increased risk of dying from cancer for people with higher serum iron [20]. Higher transferrin saturation (circulating iron) or serum iron concentration was also associated with increased risk of nonskin-cancer death in a Western Australian population [21]. 

The problem with the literature data is that there are few studies focusing on the association between iron and cancer prognosis. Thus, their results are not comparable- due to heterogeneous populations, different cohorts of patients, or methods of iron evaluation. They are also based upon a limited number of cancer cases. 

The association between iron serum levels and melanoma prognosis has not been evaluated, although lower concentrations of iron (measured in toenails) were reported among melanoma patients in one Italian study [22].

There are studies exploring prognostic gene models related to ferroptosis in tumors. Liu et al. identified the prognostic significance of ferroptosis-related genes (FRGs) in melanoma, and proposed a novel four-gene prognostic signature that may impact the assessment and clinical decision making for melanoma patients [23]. A recent study suggested a 10-biomarker signature that could be clinically used to predict the prognosis of cutaneous melanoma [24]. 

Similarly, a robust prognostic indicator based upon the evaluation of 15 FRGs was described and suggested by Zeng et al. as an independent prognostic model for melanoma in clinical application [25]. 

The evaluation of the cohort of 375 Polish MM cases revealed that a low serum iron concentration (i.e., below 893.05 μg/L) was associated with increased mortality. All study participants fasted before blood sample collection for iron level evaluation, and the examination was performed before treatment other than the surgical removal of melanoma. None of the host factors (e.g., Clark, Breslow status) was associated with iron concentrations; it is likely that the association was due to unrecognized confounding. Our study has some limitations. We had no data on BMI status. Iron was measured only once, and a single iron measurement reflected short-term iron intake. Although the patient cohort was relatively large, the small samples for various subgroups were not well-powered in our subgroup analyses. Another limitation is that the conclusions from this study may not be applicable to other populations, and we focused our analysis on serum iron, one of several biomarkers of iron status, which may or may not correlate with body iron stores. However, we saw a significant association between iron and melanoma survival in both univariable and multivariable analyses. The relationship was significant among cases with low iron concentration and tendency. An association between other factors, such as age, Breslow and Clark status, and disease outcome was found in univariable analysis. 

## 5. Conclusions

We showed that a low iron level might contribute to worse survival for patients with melanoma. Future studies based upon the examination of a larger number of cases are needed.

## Figures and Tables

**Figure 1 biomedicines-10-03018-f001:**
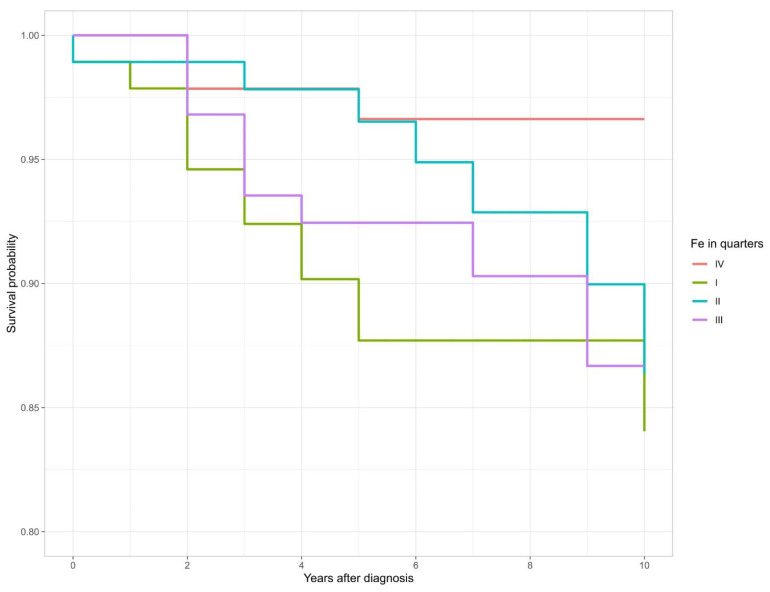
Ten-year survival by Fe level (quarters) in melanoma patients.

**Table 1 biomedicines-10-03018-t001:** Characteristics of the study group and uni- and multivariable Cox proportional hazard models for given factors (*n* = 375).

	Frequency			Univariable Cox Regression			Multivariable Cox Regression		
Characteristic	Overall, ^1^ *n* = 375	Alive, ^1^ *n* = 344	Deceased, ^1^ *n* = 31	HR ^2^	95% CI ^2^	*p*	HR	95% CI	*p*
Fe level in quarters (µg/L)									
I (162.15–893.05)	94 (25%)	82 (24%)	12 (39%)	4.12	1.16, 14.6	0.028	4.66	1.28, 16.9	0.019
II (895.69–1095.88)	93 (25%)	86 (25%)	7 (23%)	2.35	0.61, 9.10	0.2	3.40	0.85, 13.6	0.083
III (1105.49–1346.75)	94 (25%)	85 (25%)	9 (29%)	3.12	0.84, 11.5	0.088	3.71	0.99, 13.9	0.052
IV (1348.63–2815.78)	94 (25%)	91 (26%)	3 (9.7%)	—	—		—	—	
Sex									
Female	231 (62%)	217 (63%)	14 (45%)	—	—		—	—	
Male	144 (38%)	127 (37%)	17 (55%)	2.09	1.03, 4.24	0.042	1.88	0.90, 3.92	0.092
Age	21.00–90.00 (54.63)	21.00– 90.00 (53.76)	38.00 – 86.00 (64.26)	1.06	1.03, 1.09	<0.001	1.05	1.02, 1.09	<0.001
Breslow (mm) *	0.20–16.80 (1.80)	0.20–16.80 (1.71)	0.50–11.00 (3.22)	1.16	1.04, 1.29	0.008	1.11	0.96, 1.28	0.2
Clark									
II	71 (19%)	70 (20%)	1 (3.2%)	—	—		—	—	
III	157 (42%)	145 (42%)	12 (39%)	5.47	0.71, 42.1	0.10	4.35	0.56, 33.7	0.2
IV/V	147 (39%)	129 (38%)	18 (58%)	8.76	1.17, 65.7	0.035	6.84	0.91, 51.7	0.062

^1^ n (%); range (Mean); ^2^ HR = hazard ratio, CI = confidence interval; * *n* = 324.

**Table 2 biomedicines-10-03018-t002:** Fe levels by subgroups (*n* = 375).

Subgroup		*n*	Mean	SD	Median	Min	Max	Range	IQR
Sex									
	Female	231	1069.14	292.19	1044.88	162.15	1792.36	1630.21	395.5
	Male	144	1236.93	398.84	1234.28	369.67	2815.78	2446.11	526.49
Clark									
	II	71	1149.54	317.63	1156.85	460.85	2067.22	1606.37	451.18
	III	157	1107.69	318.32	1090.98	162.15	1934.12	1771.97	411.51
	IV/V	147	1153.5	386.68	1136.38	412.76	2815.78	2403.02	484.64
Fe level in quarters (µg/L)									
	I (162.15–893.05)	94	725.16	145.7	762.68	162.15	893.05	730.9	199.85
	II (895.69–1095.88)	93	999.41	59.49	998.63	895.69	1095.88	200.19	102.3
	III (1105.49–1346.75)	94	1228.17	68.56	1232.03	1105.49	1346.75	241.26	122.53
	IV (1348.63–2815.78)	94	1580.12	236.59	1529.21	1348.63	2815.78	1467.15	217.96

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
