# Peer review of "Serum Iron Level and 10-Year Survival after Melanoma"

_biomedicines, 2022, doi:10.3390/biomedicines10123018_

Round 1
Reviewer 1 Report
The manuscript presents interesting data
on survival of melanoma patients and
content of Fe in serum. There are minor points ro address.
Fig.1. Please prepare better quality Figure.
Best is to use Prism Graph Pad software.
2. Please, include in Discussion review of
opposite data, discuss differences between
groups, explain potential reasons of
different results.
3. Add some limitations of the study.
Author Response
We have responded to all the reviewer's comments in the attached PDF. Thank you for preparing the review for our publication.

Reviewer 2 Report
The authors describe the association of serum iron with prognosis of melanoma. High risk group shows poor prognosis in melanoma patients. These data are informative for management of melanoma. I have the following suggestions.
1. Ferroptosis of melanoma was described in other paper (ex. PMID: 35309911). You should discuss about ferroptosis in melanoma.
2. There are too many examples of past papers in the discussion part. Association of melanoma with iron should be more discussed.
Author Response

(The authors gave the same response as above.)
